# Understanding Consumers’ Preferences for Protected Geographical Indications: A Choice Experiment with Hungarian Sausage Consumers

**DOI:** 10.3390/foods11070997

**Published:** 2022-03-29

**Authors:** Áron Török, Matthew Gorton, Ching-Hua Yeh, Péter Czine, Péter Balogh

**Affiliations:** 1Department of Agribusiness, Corvinus University of Budapest, 1093 Budapest, Hungary; aron.torok@uni-corvinus.hu; 2Business School and National Innovation Centre for Rural Enterprise, Newcastle University, Newcastle NE1 4SE, UK; matthew.gorton@newcastle.ac.uk; 3Institute for Food and Resource Economics, University of Bonn, 53115 Bonn, Germany; chinghua.yeh@ilr.uni-bonn.de; 4Department of Statistics and Methodology, University of Debrecen, 4032 Debrecen, Hungary; balogh.peter@econ.unideb.hu

**Keywords:** protected geographical indications, private brand, taste, Hungary, processed meat, consumer preferences, stated choice experiment

## Abstract

Geographical Indications (GIs) can increase producer margins and contribute to local economic development, but the extent to which they do so depends on the nature of consumer demand. A Discrete Choice Experiment (DCE) considers the value that consumers place on a Protected Geographical Indication (PGI) in comparison with a leading manufacturer’s brand, as well as the importance of taste variations. Based on an application of DCE to sausages in Hungary, results indicate that a PGI can generate value to consumers exceeding that conveyed by the leading manufacturer’s brand. Consumers’ taste preferences, however, may not be consistent with the specification of GI products. Latent Class (LC) and Random parameter Latent Class (RLC) analyses identify two consumer segments, with the majority of consumers (71%-LC, 65%-RLC) classified as traditionalists, who most value the GI label, while a minority (29%-LC, 35%-RLC) is brand conscious, for whom the GI status is less salient. Both theoretical and business implications for GI marketing and club branding are drawn.

## 1. Introduction

Geographical Indications (GIs) reflect associations between a product and territory, which, when protected under law, prevent misuse or imitation of the registered name and guarantee consumers that the product is authentic. Famous GIs include Champagne, Parma ham, Feta cheese, and Scotch Whisky. Certified GIs are produced according to a Code of Practice which specifies the production process for the good, its distinctive qualities, and the geographic boundaries of the production area. Applicants for GI status must demonstrate that their product is “traditional” within the geographical boundaries of the production area. Only goods produced in accordance with the Code of Practice may use the GI name, restricting the outsourcing of production from the designated area [1,2,3,4]. Typically, multiple producers within the designated geographical area are part of a consortium, such that use of the protected GI name is shared between members. Consequently, GIs are club goods [5], distinguishing them from most brands which are private goods and origin labels which can be used by all entities in a particular territory.

European consumers overwhelmingly support the principles underpinning GIs. For instance, European citizens state, in their decisions to buy foods, respecting local tradition and know-how, choosing foods that come from a geographical area that they know, and having a specific label ensuring the quality of the product is either very or fairly important (82%, 81% and 82%, respectively) [6]. However, the sales of Protected Designation of Origin (PDO) and Protected Geographical Indication (PGI) certified foods, the two European Union schemes that seek to protect GIs, are more modest. GIs account for approximately 7% of the EU’s food and drink sector total sales [7]. However, when wines, spirits and other alcoholic drinks are excluded, the share of sales accounted for by PDO and PGI products is much lower—around 3%. Few PDO or PGI food products are market leaders in their product category [8].

Several factors likely explain the discrepancy between the stated importance and sales of PDO/PGI foods. While consumers, generally, value origin [9], traceability and tradition in foods, other attributes such as manufacturers’ brands convenience and taste may be more important [10]. Food is often purchased based on habit, and the influence of front-of-pack labelling on consumer decision making can be limited [11], such that the impact of PDO and PGI logos on consumers may be modest. Moreover, the higher prices usually charged for PDO/PGI goods compared to non-certified alternatives may substantially curb demand [12], especially given that the demand for PDO/PGI food is either as elastic or even more so than that for non-certified equivalents [13]. These factors can be studied through various means, including choice experiments.

Choice experiments are an established method for better understanding consumer preferences, especially where products incorporate multiple and varying search, experience and credence attributes [14]. Therefore, this approach is regularly used for exploring food related consumer attitudes (e.g., [15,16]). Consequently, choice experiments can be useful in responding to calls for a better understanding of sustainable consumption and behaviour [17]. Several previous choice experiment-based studies consider consumers’ Willingness to Pay (WTP) for PDO/PGI foods, typically evaluating GI alongside origin (local, national, imported) and organic attributes. However, previous GI-related choice experiments pay little attention to two potentially important strategies for adding value to agri-food products. First, producers can add value through their own brand name, and this may be more rewarding than a PDO/PGI designation [18], generating the question as to how much effort should be placed on individual versus club branding via a GI. A second common strategy for adding value to food products is to introduce new variants of a core product [19], which differ in taste or nutritional content (different taste variants, low sugar version etc.). The degree to which this is compatible with traditional foods, however, deserves further investigation. Traditional foods may evoke strong consumer expectations regarding the taste and composition of the product [20], with innovations regarded suspiciously [21,22]. Such reactions may limit the ability of traditional foods to appeal to growing consumer segments, who may desire more exotic, healthier, indulgent or novel tastes, trapping them as “museums of production” [23].

In responding to these gaps in the literature, this study presents a Discrete Choice Experiment (DCE) relating to sausage in Hungary, incorporating the attributes of brand/PGI, spiciness (taste variation), and price. The paper contributes to the GI and branding literature. Furthermore, methodologically, our study explores the advantages of adopting Random parameter Latent Class (RLC) models to understand heterogeneity in consumer preferences. In so doing, the paper responds to academic calls, relating to GIs, for “*research on the way in which such labels do or can affect consumer behaviour … to implement an evidence-based policy in the area*” [24] as well as policy calls to “*support and encourage research on GIs to better identify drivers of success*” with a particular focus on consumer adoption [3].

Therefore, the aim of the study is to examine the preferences of Hungarian consumers for packaged sausages, with a special focus on the PGI label.

### Willingness to Pay for GI-Labelled Foods

To analyse the extant literature on consumers’ WTP for GI-labelled foods, the authors undertook a systematic literature review, following the guidelines of Tranfield et al. [25] and Paul and Criado [26]. Following an evaluation of papers, Appendix A details 17 relevant studies. It reveals that in the last twenty years, several WTP studies measured the effects of GI labels using stated preference methods. The research draws mainly on data for Europe (Spain and Italy, in particular), with most attention given to olive oil. In general, the findings indicate a preference for GI labels [4,27]. However, to be a local (and/or national) product was as, or more, important than possessing a PDO or PGI label [28,29,30,31,32,33]. Comparing different forms of GI protection, consumers typically value PDO-labelled products more than PGI; however, the results are mixed when comparing GIs against organic [32,33,34,35,36,37,38,39]. Other product-specific attributes also matter. For Spanish lamb, the cut of meat (leg or chop) was more important than GI [40], and in the case of Italian dry-cured ham, the duration of ageing affects WTP [41].

Outside of GI-focused studies, prior research establishes that food consumers face a large amount of information on food packaging, some of which might be redundant or misleading [42], such that the influence of nutrient content claims on consumer behaviour may be modest [43]. In terms of behaviour, significant differences exist between food label users and non-users [44]. Some quality schemes (e.g., organic food) receive a growing preference and therefore attract substantial price premiums [45]. However, consumers may be confused by different labels [46] and, as evident from the systematic literature review, there is limited information on the relative importance for consumers of GI, taste and brand attributes for food products.

## 2. Methodology

### 2.1. Case Description

Meat is the predominant source of protein in Central and Eastern European diets [47]. It represents a substantial share of consumer food expenditure [48], and sausage is an important meat product in the region [49]. Taste is generally regarded as a major factor for consumers when buying sausages, even novel, functional ones [50,51].

In Hungary, chicken and pork are the best-selling meats [52], with the most popular pork meat product being dried and smoked sausage [53]. Sausage is an important constituent of Hungarian cuisine, widely used in many traditional dishes, and both spicy and non-spicy varieties are consumed daily by every second Hungarian [54]. For centuries, the typical seasonings of traditional Hungarian sausages have been salt, garlic, cumin seeds and, most importantly of all, ground (or milled) paprika. This last ingredient has been a distinctive spice of Hungarian cuisine since the 16th century [55]. Whether the ground paprika used is sweet or hot affects whether the sausage is spicy or not. Traditional sausages are spicy, and ancient recipe books often call the hot paprika used for flavouring sausages “Hungarian paprika” [56].

Gyulai sausage accounts for around 15% of total Hungarian sausage production. It is a traditional variety produced for centuries in the south-eastern part of Hungary, mainly in and around the municipality of Gyula [57]. Since 2010, it has possessed PGI status. The product is marketed by a few dominant traditional food processing companies which use the PGI label in their marketing [55]. Therefore, the DCE focused on sausage, a product which is available through all types of food retailers (e.g., supermarkets, independent grocers, food discounters, farmers’ markets). The study focused on pre-sliced and packaged sausage, which is the most popular form sold through retailers.

### 2.2. Experiment Design

After a statement of introduction and research motivation, the questionnaire began with screening questions. For inclusion, participants had to live in Hungary, had to be responsible at least partly for household food shopping, and had to have bought sausage in the last three months. This first part of the survey also collected data on respondents’ main characteristics: gender, age, location, the highest level of education and income level, together with sausage purchasing and consumption habits (frequency and price).

After conducting an internal discussion between academic researchers and market experts, and based on the literature introduced earlier in this study, three attributes were included in the survey: labels, taste and price. For labels, we provided options of no certificate (only textual description of the product), the PGI logo accompanied with the registered GI sausage name (Gyulai), and the brand name and logo of the leading brand of processed meat in Hungary (Pick Szeged Ltd.). Pick Szeged Ltd. is a traditional meat processing company, and the brand name “Pick” is well known among Hungarian consumers, with previous market research revealing that in Hungary this is the most popular processed meat brand and more than two-thirds of consumers recognize the Pick logo [53]. For taste, options of non-spicy, spicy and extra spicy were included, reflecting that spiciness is an important characteristic of the product. The four different price levels reflect actual retail unit prices for the most popular 70 g package (identified from a shop check). Table 1 illustrates the respective attributes and their levels.

The DCE formed the second part of the survey. First, the upcoming purchase simulation was explained for respondents, using a “cheap talk script” [58,59]. Three different options of sausage, together with an opt-out option, were provided. For the choice experiment, the appropriate sausage attributes and their respective levels, as well as an adequate sausage-purchasing scenario, were designed and defined.

As discussed above, the DCE included product labelling, taste, and price attributes. For the labelling and taste attributes, a dummy specification was used, and the base levels were “sausage with no certificate” and “non-spicy”. Due to the large number of possible choice sets received by full factorial design, we used a D-efficient experimental design using Ngene 1.2 software [60]. Consequently, respondents had to make six choices, in each case out of four alternatives (these always included a “would not buy any” option). Figure 1 provides an example of a decision situation.

### 2.3. Data Collection

Data collection occurred through a commercial market research agency to obtain a representative sample of food shoppers in Hungary. The research undertaken obtained ethical approval from the co-ordinating institution of the Strength2Food project prior to its commencement (Ref: P12725, Ethics and Governance screening, date: 22 May 2015).

The questionnaire was first developed in English, then translated into Hungarian, and back-translated by a professional agency, resulting in minor modifications. Finally, after pretesting, data collection occurred during the summer of 2018. The average completion time was approximately 10 min.

After filtering out incomplete responses, the analysed sample contained 380 respondents (Table 2). Compared to the Hungarian population, our sample is slightly biased towards males and consumers with fewer children, but representative for the size of the households. However, middle-aged, better-educated and urban respondents are overrepresented, which is typical for only online surveys [61].

### 2.4. Empirical Method and Model

DCEs are based on random utility theory assuming that individuals always choose the option from a decision set that provides them with the highest level of utility, and only a certain part of this utility can be observed by the researcher. Thus, the total utility can be separated into a systematic and a random part according to Equation (1) [63,64].
(1)$$U_{n,i,t}=V_{n,i,t}+\varepsilon_{n,i,t}$$
where *U* is the total utility, *V* is the systematic part, *ε* is the random part, *n* is the respondent, *i* is the alternative, and *t* is the decision situation.

For the models, the systematic part of utility is depicted by Equation (2) in our case.
(2)$$V_{i}=ASC_{alt.\;\;No\;choice}+\beta_{Price}Price_{alt.\;\;i}+\beta_{GI\;label}GI\;label_{alt.\;\;i}+\beta_{Private\;brand}Private\;brand_{alt.\;\;i}\;+\beta_{Spicy}Spicy_{alt.\;\;i}+\beta_{Extra\;spicy}Extra\;spicy_{alt.\;\;i}$$
where $V_{i}$ is the systematic part of the utility for the *i*-th alternative; *β* is the vector of parameters estimated for the attributes; $ASC_{alt.\;No\;choice}$ is the alternative specific constant for no choice; *Price*, *GI label*, *Private brand*, *Spicy* and *Extra spicy* are the attributes in our experiment.

The literature employs several model specifications to analyse data obtained from a DCE, one of the oldest being the multinomial logit model (MNL) [64]. The latter’s advantages include that it is relatively easy to estimate, and results are easy to interpret. However, it also presents some disadvantages, including assumptions regarding homogeneous consumer preferences.

To handle heterogeneity in consumer preferences, we can introduce discrete or/and continuous distributions during the model building. When we use only discrete distributions, we can estimate a latent class (LC) model. The latter forms distinct heterogeneous classes within which members’ preferences are homogeneous [65]. The limitations of the LC model, however, include difficulties determining the appropriate number of classes, which are mostly decided on the basis of information criterion, namely: Akaike information criterion (AIC), and Bayesian information criterion (BIC) [66].

An important feature of the LC model is that it calculates class allocation probabilities, through which it is possible to predict the probability that individuals fall into certain classes. In our case, we use a class allocation equation with only a constant according to Equation (3).
(3)$$C_{Allocation_{n,q}}=\delta_{q}$$
where $C_{Allocation_{n,q}}$ is the probability of the *n*-th respondent fall into the *q*-th class, and $\delta_{q}$ is the constant for the *q*-th class (one is fixed at 0, only a constant of *Q*-1 is estimated). Another modelling approach allows for accommodating preference heterogeneity through the use of continuous distributions. The mixed logit model (MLM) allows the *β* parameters to vary among respondents according to a predetermined distribution (by the researcher) and then estimating certain parameters (expected value, standard deviation) [67].

One extension of the above-mentioned two models (LC and MLM) is the random parameter latent class (RLC) model, which captures heterogeneity in preferences from two directions. It forms distinct groups in a similar way to the LC model, but it also allows continuous random heterogeneity for parameters [68,69].

Willingness to Pay (WTP) was also estimated for all models, using the WTP-space approach according to Equation (4) [70].
(4)$$V_{i}=ASC_{alt.\;\;No\;choice}+\beta_{Price}(Price_{alt.\;\;i}+{\mathrm{WTP}}_{GI\;label}GI\;label_{alt.\;\;i}+{\mathrm{WTP}}_{Private\;brand}Private\;brand_{alt.\;\;i}\;+{\mathrm{WTP}}_{Spicy}Spicy_{alt.\;\;i}+{\mathrm{WTP}}_{Extra\;spicy}Extra\;spicy_{alt.\;\;i}$$
where *WTP* is the vector of WTP-s estimated for the attributes.

The Apollo 0.2.4 R package was used to estimate the models [71,72,73].

## 3. Results

### 3.1. Buying and Consumption Habits

Overall, 60% of those surveyed were fully responsible for their household’s food purchases, while the remainder only partially performed this task. All respondents had bought sausages in the last three months, for which the majority (26.58%) paid between 291 and 360 HUF (c.a. EUR 0.91–1.13) for a package of 70 g sliced sausage. In terms of the frequency of sausage purchases, the largest share (27.89%) was made up of those who bought fortnightly, while consumption was typically two or three times a week (37.10%). Detailed data are shown in Table 3. Overall, our sample shows similar characteristics to previous market research for Hungary (e.g., [53,54]) as our respondents buy sausage regularly (almost 93% at least once a month) and consume it very often (72% at least once a week).

### 3.2. Model Estimations

Table 4 presents the MNL and MLM estimates. In the latter case, all parameters were included as random, as we obtained significant standard deviations for all attributes [14]. A lognormal distribution was used for price and a normal distribution for all other attributes. Our estimates were made with 500 Modified Latin Hypercube Sampling (MLHS) draws [74]. These two models and the models estimated subsequently were also based on the utility function detailed in the empirical method and model part.

Based on the estimates of the MNL model, it is evident that the “opt-out” (not buying) option is less preferred by respondents compared to the choice of a sausage alternative. The negative value of the price coefficient, as one would expect, suggests that an increase in the price reduces consumers’ sense of utility. Consumers prefer the GI-labelled product to the non-labelled, and to a lesser extent, the private branded product. As the spice content of the product increases, consumers’ sense of utility decreases.

Compared to the MNL model, the MLM shows a substantially better fit, based on information criteria (log-likelihood, Pseudo $R^{2}$, AIC, BIC). Furthermore, the significant standard deviations obtained for the attributes suggest that there is heterogeneity in consumer preferences that the MNL model cannot accommodate. Based on the alternative-specific constant (which was not included randomly) and parameter estimates for attributes obtained in the case of the MLM, we can draw similar conclusions as for the MNL model. Consumers most prefer the GI-labelled product with no further spice at the lowest possible price.

The LC specification provides an opportunity to separate groups with similar preferences within our sample. An extension of this, the RLC model also provides an option to address not only heterogeneity among groups, but also randomness within groups that is not handled by grouping. Estimates of the two-class LC and RLC models are shown in Table 5. While versions with more than two classes were also considered, they were rejected as the parameters of the RLC model no longer showed realistic results [68]. Consequently, the two-class version was selected for both models for comparability. In the case of the RLC model, such as the MLM, all parameters were presented as random, with a lognormal distribution for price and a normal distribution for the additional attributes with 500 MLHS draws.

The negative value for the “no choice” option in the LC/RLC results indicates that the sausage options presented were generally preferred to not buying. Concerning the two-class LC model, we can conclude that the first group (class 1) is more populous (based on the positive value of δ). They prefer the GI-labelled product and have a positive attitude towards spicy sausage. Respondents are less likely to belong to the second group (class 2), who are more price sensitive, prefer private branded products, and reject both spicy and extra spicy varieties. The RLC model also identifies these groups with a significantly better fit. However, several differences can be observed. In the first group, heterogeneity can be identified for all attributes (except the attribute of spiciness), while in the second group, heterogeneity cannot be identified for the attributes of private brand and extra spiciness. Finally, for the first group, neither the LC nor the RLC model estimates indicate a significant effect for the extra spicy attribute.

### 3.3. Willingness to Pay (WTP) Estimates

The WTP estimates for the four models presented above are shown in Table 6, which were based on the formula presented in Equation (4).

Based on the results of WTP estimates, we can conclude that labels (both GI and private brand) always result in a positive willingness to pay. Consumers in class 1 are ready to pay the highest price for the GI label, of approximately 107 to 148 HUF (c.a. EUR 0.33–0.46), while consumers in class 2 are willing to pay the most for the private brand, approximately 120 to 158 HUF (c.a. EUR 0.38–0.48).

In our models, taste preferences regarding spiciness clearly separate consumers. Both MNL and MLM models suggest that consumers are willing to pay less for a spicy product compared to the non-spicy alternatives (they would pay less, approximately 61 and 107 HUF (c.a. EUR 0.17–0.29) for spicy, and approximately 161 to 224 HUF less (c.a. EUR 0.44–0.61) for extra-spicy sausage). Based on the groupings of the LC models, it is apparent that consumers in class 1 prefer spicy sausage and would pay more, approximately 67 to 116 HUF (c.a. EUR 0.18–0.32). As for the consumers in class 2, their rejection of spiciness is also manifested in their WTP, as they would pay less for these products, approximately 484 to 619 HUF (c.a. EUR 1.32–1.69), and would pay even less for the extra-spicy sausage, approximately 590 to 869 HUF (c.a. EUR 1.61–2.37).

Based on their characteristics, we label class 1 as *Traditional Consumers* and class 2 as *Brand Conscious Consumers.* Descriptive statistics for the two groups in terms of sociodemographic factors are shown in Table 7.

It is possible to consider the degree of similarities and differences between members of the two classes based on their purchase behaviour, reported in the survey. Table 8 reveals a lack of significant differences between traditional and brand conscious consumers regarding the average prices paid for sausages. However, the frequency of consumption differs significantly between the two groups. Traditional consumers consumed the product more frequently (often twice or three times a week) than the brand conscious group, who were more likely to be weekly or monthly consumers.

## 4. Discussion

### 4.1. Theoretical Implications

Geographical indications have become increasingly prominent in regional and local economic policy, theoretically offering the possibility of higher returns to producers while also benefiting workers and the wider communities in which they are based [75,76]. However, the socio-economic effects of GIs depend in part on consumers’ willingness to pay for them [32]. A better understanding of the nature of consumer demand for GIs is thus warranted.

Brands have been traditionally conceptualised as the unique assets of enterprises, with brand building planned and executed by their unique owners [77,78]. However, GIs are club goods [5], shared by members of the producers’ consortium but unavailable non-members, distinguishing them from public goods and generic origin and “made in” labels [79]. The extant research pays scant attention to the value of club brands, and following calls for increased consideration of alternative brand ownership models [80], this paper contributes through an evaluation of the relative merits of club versus individual branding. We find that in the Hungarian case, the club group brand (PGI) generates superior value in comparison to the leading manufacturer’s brand. This result gives credence to calls to investigate non-private good brands and alternative architectures for generating brand value [81].

When considering the nature of consumer demand within a particular product category, it is important not to overgeneralise but rather capture the heterogeneity of consumers, detailing relevant consumer segments. Much of the extant literature on the consumer appeal of GIs rests on a discussion of general sentiments and average ratings [82], including for WTP [4]. However, the latter approach fails to adequately understand consumer segments. According to the LC and RLC models, two distinctive groups of consumers are apparent in our case. Based on their characteristics, we named them *Traditional* and *Brand Conscious Consumers*. The first group was larger, accounting for 71% of respondents, pointing to the potential mass market appeal of GI products. The second group, for whom the private brand was more salient, had a higher proportion of women consumers. It is important to underline that for both groups, labels were important, however, to varying degrees. For *Traditional Consumers,* the PGI label was strongly preferred, indicating that for this group, the club brand is more important than that of individual producers.

One criticism of GIs is that they can become “museums of production” [23], embodying a static notion of what a particular product should be, which becomes increasingly out of touch with consumer tastes and demand. Some concerns of this nature were apparent regarding the spiciness attribute. Hungarian sausage is traditionally spicy, but many consumers prefer a milder flavour. This was particularly evident in the WTP estimates for *Brand Conscious Consumers* for whom spiciness has a substantial disutility, and WTP was greater for the private branded version (compared with PGI). However, such consumers were in the minority, and in the Hungarian case, there was thus little evidence that traditional foods will lose their appeal in the future.

### 4.2. Business Implications

The food industry generally is a low margin business [83], with managers searching for strategies to increase the added value of their products. Two important strategies for adding value are through branding and product modification, such as through adjusting the ingredients to alter consumers’ sensory experience. Evaluating the relative merits of these different strategies for adding value is an important practical challenge.

The DCE results offer reassurance to managers regarding the potential consumer appeal of GIs, and PGIs in particular. Overall, consumer WTP for the PGI product matched or exceeded that for the leading private branded product. In particular markets, club brands can thus generate mark ups equivalent to or better than individual branding. This is consistent with other evidence that traditional food products can command a substantial price premium, especially once they are backed by an official designation such as the PGI label [84]. The appeal of a PGI product may not be limited to a particular age or income group, and in Hungary, GI products benefited from recent government-funded campaigns aiming at increasing consumers’ awareness of traditional meat products [85]. Moreover, the results support the notion that consumers care about GI labels [86], and at least in Hungary, they affect WTP substantially. Using a GI label in marketing can have a positive influence on producers’ margins.

Critical to the success of a GI-based branding strategy will be the extent to which consumers’ taste preferences are consistent with the version of the product enshrined in the GI’s Code of Practice. Managers should thus assess this degree of consistency as part of their audit of the market environment before developing their strategy. In the Hungarian case, most consumers’ taste preferences were consistent with the GI’s composition, albeit with a sizeable minority preferring a less spicy option. However, this may not always be the case, and a GI-based business strategy will be far less appealing when consumers’ taste preferences are inconsistent with traditional formulations.

### 4.3. Limitations and Future Research Directions

While generating insights regarding consumers’ preferences for GIs and the relative merits of club versus individual branding, this study is not without limitations. To avoid consumer confusion and cognitive overload, it considered a small number of critical attributes. However, for further insight, future research could investigate additional attributes (e.g., differences between types of retail outlets) and apply extended methodological approaches (e.g., hybrid choice modelling), together with considering other certifications (e.g., measuring differences between PDO and PGI labels). Regarding the latter, survey evidence suggests that consumers typically lack a good understanding of the differences between PDO and PGI designations [10], and it would be useful to see if this is reflected in WTP. Finally, this study investigates the appeal of a GI product for local consumers where it is a part of the traditional cuisine. It would be interesting to investigate if and how WTP for GIs varies in export markets, and to investigate the effectiveness of strategies for enhancing a GI’s reputation in markets where it is currently little known. The latter is important for the use of GIs as an economic development tool, particularly for territories where the local market is small and with limited purchasing power [87].

## 5. Conclusions

Our study utilized a choice experiment to assess Hungarian consumers’ willingness to pay for different labels (geographical indications and private brand) and variations in spiciness considering a package of 70 g smoked and dried sausage. The study details that the PGI label can generate a substantial price premium, comparable to or exceeding that of the leading private brand. Latent Class (LC) and Random parameter Latent Class (RLC) analyses identified two consumer segments, with the majority of consumers (71%-LC, 65%-RLC) classified as traditionalists, who most value the GI label, while a minority (29%-LC, 35%-RLC) is brand conscious. Most consumers in Hungary, therefore, fall into a traditional sausage segment that is well disposed to GI labelling. In contrast, a smaller segment prefers the private brand, strongly rejects spiciness, and has a higher proportion of women.

Our findings contribute to the existing literature on consumers’ willingness to pay for GI-labelled products, using stated preference methods, with special attention given to branding and taste as a sensory attribute.

## Figures and Tables

**Figure 1 foods-11-00997-f001:**
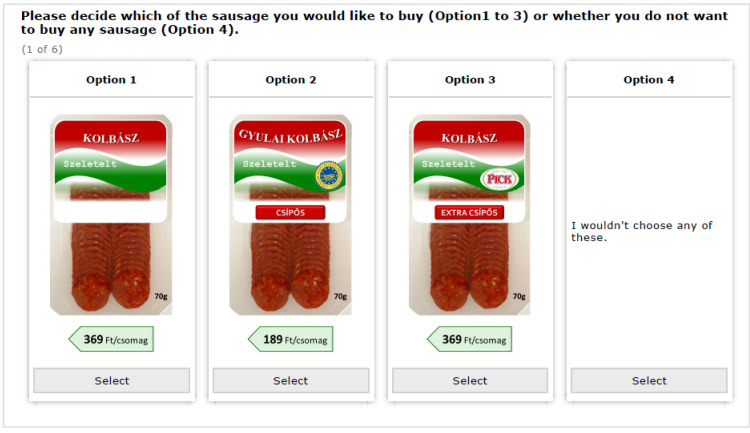
An example of a decision situation. Note: Option 1: sliced (‘Szeletelt’) sausage (‘Kolbász’), 369 HUF for a 70 g package (‘369 Ft/csomag’); Option 2: sliced Gyulai sausage (‘Gyulai kolbász’) with PGI label, spicy (‘Csípős’), 189 HUF for a 70 g package; Option 3: sliced sausage with private brand label Pick, extra spicy (‘Extra csípős’), 369 HUF for a 70 g package.

**Table 1 foods-11-00997-t001:** Attributes and respective levels used in the Discrete Choice Experiment.

Attributes and Respective Levels
**Labels:** Sausage with no certificateGyulai sausage PGI 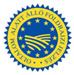 PICK sausage 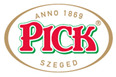 **Taste** Non-spicySpicyExtra spicy **Price * (HUF/ 70 g):** 189 HUF (circa 0.59 EUR)279 HUF (circa 0.87 EUR)369 HUF (circa 1.15 EUR)459 HUF (circa 1.43 EUR)

* In the analysis, we applied an exchange rate of 320 HUF/EUR as an average rate at the time of data collection (summer of 2018).

**Table 2 foods-11-00997-t002:** Sociodemographic characteristics of respondents.

Sociodemographic Factors	Sample (*N* = 380)	Hungarian Population *
Gender (%)		
Female	49.47	52.15
Male	50.53	47.85
Age (category) (%)		
<30	23.16	32.81
30–39	21.58	11.75
40–49	24.21	16.25
49<	31.05	39.19
Highest level of education (%)		
Upper secondary/lower secondary/primary education or below University or college entrance qualification Bachelor’s, master’s or doctoral degree	31.32	51.83
	25.52	29.45
	43.16	18.72
Monthly net income (%)		
<150,000 HUF (<c.a. EUR 469)	6.32	244,609 HUF (c.a. EUR 764)/month
150,000–205,000 HUF (c.a. 469–EUR 641)	11.58	
205,001–235,000 HUF (c.a. 64–EUR 734)	11.58	
235,001–380,000 HUF (c.a. 734–EUR 1188)	38.68	
380,001–835,000 HUF (1188–EUR 2609)	30.00	
835,000 < HUF (c.a. EUR 2609<)	1.84	
Residence (%)		
City	46.84	37.91
Urban (non-cities)	36.32	32.58
Rural	16.84	29.51
Household size (mean)	2.86	2.86
Number of children (<18 years) in a household (mean)	0.57	1.06

Note: * Hungarian Central Statistical Office [62].

**Table 3 foods-11-00997-t003:** Buying and consuming habits of respondents.

Average Price Normally Paid for a 70 g Package of Sausage (%)	
Below 150 HUF (c.a. EUR 0.47)	1.32
Between 150–220 HUF (c.a. EUR 0.47–0.69)	13.68
Between 221–290 HUF (c.a. EUR 0.69–0.91)	17.90
Between 291–360 HUF (c.a. EUR 0.91–1.13)	26.58
Between 361–430 HUF (c.a. EUR 1.13–1.34)	15.26
Between 431–500 HUF (c.a. EUR 1.34–1.56)	8.16
Above 500 HUF (c.a. EUR 1.56)	7.63
Does not know	9.47
Frequency of purchase (%)	
Less than once a month	7.11
Once a month	25.53
Twice a month	27.89
Three times a month	15.26
Once a week	20.79
More than once a week	3.16
Does not know	0.26
Frequency of consumption (%)	
Less than once a month	8.42
Twice or three times a month	18.42
Once a week	26.32
Twice a week, three times a week	37.10
Four to six times a week	7.11
Every day	1.58
Does not know	1.05

**Table 4 foods-11-00997-t004:** Coefficient estimates by multinomial logit and mixed logit models.

Attributes and Model Details	MNL		MLM	
	Coeff.	S.E.	Coeff.	S.E.
ASC no choice	−2.39 ***	0.11	−4.41 ***	0.22
Price (scaled by 100)	−0.43 ***	0.02	−0.86 ***	0.07
Price (SD)	−	−	0.77 ***	0.11
GI label	0.58 ***	0.06	0.95 ***	0.11
GI label (SD)	−	−	0.82 ***	0.15
Private brand	0.53 ***	0.06	0.90 ***	0.10
Private brand (SD)	−	−	0.85 ***	0.14
Spicy	−0.26 ***	0.06	−0.51 ***	0.14
Spicy (SD)	-	-	2.09 ***	0.17
Extra Spicy	−0.69 ***	0.06	−1.35 ***	0.17
Extra Spicy (SD)	-	-	2.39 ***	0.19
$\mathrm{Pseudo}\;R^{2}$	0.15		0.28	
Log-likelihood (0)	−3160.75		−3160.75	
Log-likelihood (model)	−2694.89		−2264.75	
AIC	5401.77		4551.50	
BIC	5436.16		4614.55	

Note: S.E. denotes the standard errors; S.D. denotes the standard deviations; ASC represents the alternative-specific constant; *** indicate statistical significance at the 1% level; AIC denotes the Akaike information criterion; BIC denotes the Bayesian information criterion.

**Table 5 foods-11-00997-t005:** Coefficient estimates by two classes latent class and random parameter latent class models.

Attributes and Model Details	LC				RLC			
	Coeff.		S.E.		Coeff.		S.E.	
ASC no choice	−3.13 ***		0.16		−4.71 ***		0.26	
	**Class 1**		**Class 2**		**Class 1**		**Class 2**	
	**Coeff.**	**S.E.**	**Coeff.**	**S.E.**	**Coeff.**	**S.E.**	**Coeff.**	**S.E.**
Price (scaled by 100)	−0.49 ***	0.03	−0.70 ***	0.06	−0.89 ***	0.08	−0.94 ***	0.12
Price (SD)	-	-	-	-	0.84 ***	0.16	0.88 ***	0.22
GI label	0.73 ***	0.08	0.39 **	0.19	0.96 ***	0.13	0.73 ***	0.27
GI label (SD)	-	-	-	-	0.91 ***	0.17	0.84 **	0.41
Private brand	0.64 ***	0.07	0.83 ***	0.22	0.86 ***	0.12	1.10 ***	0.19
Private brand (SD)	-	-	-	-	0.95 ***	0.15	0.06	0.40
Spicy	0.57 ***	0.09	−3.37 ***	0.28	0.79 ***	0.13	−3.93 ***	0.62
Spicy (SD)	-	-	-	-	0.29	0.34	3.85 ***	0.64
Extra Spicy	0.06	0.09	−4.11 ***	0.40	0.24	0.20	−4.35 ***	0.43
Extra Spicy (SD)	-	-	-	-	1.30 ***	0.18	0.40	0.82
δ	0.90 ***	0.13					−0.61 ***	0.16
Class probability values	0.71		0.29		0.65		0.35	
$\mathrm{Pseudo}\;R^{2}$	0.24				0.31			
Log-likelihood (0)	−3160.75				−3160.75			
Log-likelihood (model)	−2401.92				−2167.03			
AIC	4827.85				4378.06			
BIC	4896.63				4504.16			

Note: S.E. denotes the standard errors; S.D. denotes the standard deviations; δ is a constant in the class allocation equation in case of the latent class models; ASC represents the alternative-specific constant; ** indicate statistical significance at the 5% level; *** indicate statistical significance at the 1% level.

**Table 6 foods-11-00997-t006:** Willingness To Pay estimates for the models.

Product Attributes	Willingness to Pay					
	MNL	MLM	LC		RLC	
			Class 1	Class 2	Class 1	Class 2
GI label	1.34 ***	1.13 ***	1.48 ***	0.57 **	1.07 ***	0.93 ***
		(1.22 ***)			(1.20 ***)	(0.27)
Private brand	1.23 ***	1.14 ***	1.31 ***	1.20 ***	0.97 ***	1.58 ***
		(0.99 ***)			(1.17 ***)	(0.24)
Spicy	−0.61 ***	−1.07 ***	1.16 ***	−4.84 ***	0.67 ***	−6.19 ***
		(2.65 ***)			(0.71 ***)	(6.58 ***)
Extra Spicy	−1.61 ***	−2.24 ***	0.12	−5.90 ***	0.28	−8.69 ***
		(3.24 ***)			(1.53 ***)	(0.09)

Note: ** indicate statistical significance at the 5% level; *** indicate statistical significance at the 1% level; The standard deviations in the MLM and RLC models are shown in parentheses below the WTP estimates.

**Table 7 foods-11-00997-t007:** Sociodemographic characteristics of classes based on the latent class model.

Sociodemographic Factors	Traditional Consumers 71%	Brand Conscious Consumers 29%
Gender (%) ***		
Female	45.86	58.39
Male	54.14	41.61
Age (category) (%)		
<30	23.85	21.44
30–39	20.37	24.57
40–49	26.26	19.15
49<	29.52	34.84
Highest level of education (%)		
Upper secondary/lower secondary/primary education Below University or college entrance qualification Bachelor’s, master’s or doctoral degree	31.48	30.91
	23.69	30.07
	44.83	39.02
Monthly net income (%)		
<150,000 HUF (<c.a. EUR 469)	6.30	6.37
150,000–205,000 HUF (c.a. EUR 469–641)	11.16	12.61
205,001–235,000 HUF (c.a. EUR 641–734)	12.86	8.42
235,001–380,000 HUF (c.a. EUR 734–1188)	36.87	43.17
380,001–835,000 HUF (EUR 1188–2609)	30.82	27.96
835,000 < HUF (c.a. EUR 2609<)	1.99	1.47
Residence (%)		
City	45.80	49.43
Urban (non-cities)	36.98	34.68
Rural	17.22	15.89
Household size (mean)	2.85	2.87
Number of children (<18 year) in a household (mean)	0.54	0.63

Note: *** indicates statistically significant difference at the 1% level, using $Chi^{2}$ test.

**Table 8 foods-11-00997-t008:** Purchasing and consumption habits of classes based on the latent class model.

Questions	Traditional Consumers 71%	Brand Conscious Consumers 29%
Average price normally paid for a 70 g package of sausage (%)		
Below 150 HUF (c.a. EUR 0.47)	1.81	0.08
Between 151–220 HUF (c.a. EUR 0.47–0.69)	11.89	18.13
Between 221–290 HUF (c.a. EUR 0.69–0.91)	17.58	18.67
Between 291–360 HUF (c.a. EUR 0.91–1.13)	27.20	25.04
Between 361–430 HUF (c.a. EUR 1.13–1.34)	15.12	15.61
Between 431–500 HUF (c.a. EUR 1.34–1.56)	9.02	6.04
Above 501 HUF (c.a. EUR 1.56)	8.74	4.90
Does not know	8.64	11.53
Frequency of purchase (%)		
Less than once a month	5.40	11.30
Once a month	23.73	29.96
Twice a month	28.51	26.37
Three times a month	15.36	15.04
Once a week	22.97	15.40
More than once a week	4.02	1.02
Do not know	<0.01	0.91
Frequency of consumption (%) **		
Less than once a month	5.61	15.36
Two-three times a month	15.70	25.16
Once a week	27.57	23.22
Two-three times a week	41.38	26.54
Four to six times a week	7.44	6.28
Every day	1.93	0.70
Do not know	0.37	2.74

Note: ** indicates statistically significant difference at 5% level, using $Chi^{2}$ test.

## Data Availability

The data presented in this study are available on request from the corresponding author.

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
