# Peer review of "Understanding Consumers’ Preferences for Protected Geographical Indications: A Choice Experiment with Hungarian Sausage Consumers"

_foods, 2022, doi:10.3390/foods11070997_

Round 1
Reviewer 1 Report
The article is very well written and easy to read along all the sections. The authors well explained the limitations and further developments/research lines at the end of the paper. I find it unfortunate that the authors chose to consider only 2 attributes in the choice, although I understand their specific focus on GIs. Below you can find some very minor points/comments that I believe would help or improve further an already good article.
- line 49: the citation "European Commission, 2020b) has to be included within the references (add the number);
- lines 157-159: add more info to justify the choice of those attributes;
- lines 162-163: add market data to justify it;
- it would be good to standardize the wording "private brand, trademark and manufacturer brand" throughout the paper;
Author Response
Dear Reviewer,
Please find attached our cover letter.
Kind regards,
The Authors

Reviewer 2 Report
The article presents an important issue from both a global (sales, cultural identity) and individual (preferences, WTP) perspective. Overall, the planning and execution of the experiment is positive. However, I have various remarks concerning the way of its description. First of all, the text needs to be reorganized, including modification of the order of presented content or more synthetic descriptions.
So there is a need to make some changes in the article.
Only Protected Geographical Indication was included in the study, so this should also be reflected in the title rather than GI.
Abstract needs to be rewritten.
Line 21. Based on an application … of what?
Line 27. Implications for GI marketing and club branding are drawn – what kind of implications – please describe them synthetically.
The authors formulated too many keywords, also their form should be reconsidered, e.g. whether heterogeneity in preferences can be replaced by consumer preferences. Moreover, abbreviation as a keyword is not a good solution.
Introduction
The statement “Geographical Indications (GIs) are associations between a product and territory” is not correct. Rather than "are" it should be "reflect".
Line 45. The sentence “European consumers overwhelmingly support the principles underpinning GIs” needs to be supported by references.
Line 96. I suggest including section 2 Background in the first chapter. Right now the introduction is longer and introduces more elements than the Background section, which only deals with WTP.
Methodology
Lines 122-140. In this section there is no need to describe the meat situation in Central and Eastern Europe. It is sufficient only to characterize the product that is the subject of the study..
Lines 143-146. Product description (presliced and packaged sausage) should be included in the previous section.
Lines 153-159. This section needs to be organized as first selection of attributes, and then about survey implementation.
Lines 173-174. Please provide more information on the questions used to assess sausage-related perceptions and purchase habits (examples of questions and answers), as well as the importance of motives underlying food choices (which motives were considered).
Lines 175-178. Rather, this information should be included in the 3.3. Data collection.
Results
Lines 254-25. This sentence should be omitted.
Line 268. Table 3. “Buying and consuming habits of respondents” should have a changed graphic form. In this form it is difficult to figure out what is going on. A more traditional table form is desirable.
Table 8 doesn't have to be that big, just taper the last two columns.
Line 360-362. This is a discussion rather than a result, so it should not be in this chapter.
The study was about PGIs, so I suggest being more careful in making various generalizations, such as those in section 5.2 Business implications. Rather, PGIs should be used instead of GIs, because that is what the study was about.
Author Response

(The authors gave the same response as above.)

Round 2
Reviewer 2 Report
I accept the changes made by Authors. There is only a need to add the aim of the study at the end of The Introduction section.
Author Response
Dear Reviewer,
Thank you for the positive feedback. We have added a short sentence to the end of the Introduction, indicating the aim of the study.
"Therefore, the aim of the study is to examine the preferences of Hungarian consumers for packaged sausages, with a special focus on the PGI label."
Kind regards,
The Authors